# Epigenetic Biomarkers and the Wnt/β-Catenin Pathway in *Opisthorchis viverrini*-associated Cholangiocarcinoma: A Scoping Review on Therapeutic Opportunities

Alok Kafle[1,2], Sutas Suttiprapa[1,2]*, Mubarak Muhammad[3], Jan Clyden B. Tenorio[1,2], Roshan Kumar Mahato[4], Norhidayu Sahimin[5], Shih Keng Loong[5]

1 Department of Tropical Medicine, Faculty of Medicine, Khon Kaen University, Khon Kaen, Thailand, 2 WHO Collaborating Centre for Research and Control of Opisthorchiasis, Khon Kaen University, Khon Kaen, Thailand, 3 Department of Physiology and Graduate School, Faculty of Medicine, Khon Kaen University, Khon Kaen, Thailand, 4 Faculty of Public Health, Khon Kaen University, Khon Kaen, Thailand, 5 Tropical Infectious Diseases Research & Education Centre, Universiti Malaya, Kuala Lumpur, Malaysia

* sutasu@kku.ac.th

## Abstract

### Background

Epigenetic modifications, such as DNA methylation and histone modifications, are pivotal in regulating gene expression pathways related to inflammation and cancer. While there is substantial research on epigenetic markers in cholangiocarcinoma (CCA), *Opisthorchis viverrini*-induced cholangiocarcinoma (*Ov*-CCA) is overlooked as a neglected tropical disease (NTD) with limited representation in the literature. Considering the distinct etiological agent, pathogenic mechanisms, and pathological manifestations, epigenetic research plays a pivotal role in uncovering markers and potential targets related to the cancer-promoting and morbidity-inducing liver fluke parasite prevalent in the Great Mekong Subregion (GMS). Emerging studies highlight a predominant hypermethylation phenotype in *Opisthorchis viverrini* (*O. viverrini*) tumor tissues, underscoring the significance of abnormal DNA methylation and histone modifications in genes and their promoters as reliable targets for *Ov*-CCA.

### Principal findings

Relevant published literature was identified by searching major electronic databases using targeted search queries. This process retrieved a total of 81 peer-reviewed research articles deemed eligible for inclusion, as they partially or fully met the pre-defined selection criteria. These eligible articles underwent a qualitative synthesis and were included in the scoping review. Within these, 11 studies specifically explored *Ov*-CCA tissues to investigate potential epigenetic biomarkers and therapeutic targets. This subset of 11 articles provided a foundation for exploring the applications of epigenetics-based therapies and biomarkers for *Ov*-CCA. These articles delved into various epigenetic modifications, including DNA methylation and histone modifications, and examined genes with aberrant epigenetic changes linked to deregulated signalling pathways in *Ov*-CCA progression.

**Data Availability Statement:** The data supporting the findings of this study are available within the manuscript and its supplementary materials.

**Funding:** NS and SKL received fundings from the Ministry of Higher Education, Malaysia under Dana Langganan SUKUK Pakej Rangsangan Ekonomi Prihatin Rakyat (SUKUK PRIHATIN)-Fasa 2 (MO002-2021). AK is supported by the Postgraduate Scholarships for International Students (PGIS), Faculty of Medicine, Khon Kaen University. SS was supported by the Fundamental Fund of Khon Kaen University and the National Science, Research, and Innovation Fund. The funders had no role in study design, data collection and analysis, decision to publish, or preparation of the manuscript.

**Competing interests:** The authors have declared that no competing interests exist.

## Conclusions

This review identified epigenetic changes and Wnt/β-catenin pathway deregulation as key drivers in *Ov*-CCA pathogenesis. Promoter hypermethylation of specific genes suggests potential diagnostic biomarkers and dysregulation of Wnt/β-catenin-modulating genes contributes to pathway activation in *Ov*-CCA progression. Reversible epigenetic changes offer opportunities for dynamic disease monitoring and targeted interventions. Therefore, this study underscores the importance of these epigenetic modifications in *Ov*-CCA development, suggesting novel therapeutic targets within disrupted signalling networks. However, additional validation is crucial for translating these novel insights into clinically applicable strategies, enhancing personalised *Ov*-CCA management approaches.

## Author summary

This review examines the role of epigenetic changes and the Wnt/β-catenin signalling pathway in *Opisthorchis viverrini*-induced cholangiocarcinoma (*Ov*-CCA). Through an exhaustive analysis of relevant studies, the primary epigenetic alteration identified in *Ov*-CCA pathogenesis is hypermethylation of tumor suppressor gene promoters. This epigenetic dysregulation shows several affected genes involved in the Wnt/β-catenin signalling pathway, which plays a central role in opisthorchiasis progression. Network analysis revealed an interaction between these genes and β-catenin, suggesting Wnt/β-catenin pathway dysregulation contributes to *Ov*-CCA development and progression. Most methylated genes act as tumor suppressors by normally inhibiting the Wnt/β-catenin pathway. Targeting this pathway offers therapeutic potential for *Ov*-CCA. Epigenetic markers hold promise as biomarkers, emphasizing the therapeutic potential of targeting disrupted gene and signalling networks through epigenetic modulators. This approach presents an innovative strategy for developing personalized therapies tailored to the molecular profiles of *Ov*-CCA patients. However, translating these findings into clinical practice requires rigorous validation and further in-depth exploration. Nonetheless, the insights obtained highlight the crucial necessity for ongoing research on epigenetic-based interventions, intending to enhance both diagnostic accuracy and therapeutic effectiveness in managing *Ov*-CCA.

## 1 Introduction

Liver fluke infections caused by parasites like *Clonorchis sinensis*, *Opisthorchis felineus*, and *Opisthorchis viverrini* (*O. viverrini*) are significant public health challenges prevalent in Eastern Europe, East Asia, and Southeast Asia [1]. *O. viverrini* infections are particularly endemic in Southeast Asian nations, including Thailand, Lao PDR, Cambodia, and Vietnam [2]. The association between liver fluke infestation and cholangiocarcinoma (CCA) is well documented. The International Agency for Research on Cancer (IARC) has classified *O. viverrini* and *C. sinensis* as Group I agents, recognising them as biological carcinogens [3].

Approximately 10 million individuals in the Great Mekong Subregion (GMS) are believed to be infected with *O. viverrini*. Over 6 million Thai residents are affected by *O. viverrini*, with the most significant prevalence in Northeastern Thailand [4]. Liver and bile duct cancer is also one of the top ten diseases associated with high mortality within the Thai population [5]. The

elevated occurrence of this concerning issue is intricately linked to the widespread existence of the liver fluke parasite, *O. viverrini*, thereby substantially heightening the susceptibility to CCA. Unfortunately, many patients in this region are diagnosed with advanced-stage CCA, resulting in an exceptionally high mortality rate [6]. Epigenetic alterations can help provide pivotal insights for early marker discovery, identify altered signalling pathways, and provide alternatives to exploring therapeutic targets.

Over the last twenty years, there has been a recognition of the critical role played by epigenetic mechanisms as key regulators capable of initiating and sustaining cholangiocarcinogenesis [7,8]. However, the understanding of the molecular genetics and epigenetic mechanisms in *Opisthorchis viverrini*-induced cholangiocarcinoma (*Ov*-CCA) is limited compared to non-*Opisthorchis viverrini*-induced cholangiocarcinoma (non-*Ov*-CCA). Exome sequencing revealed evident differences in mutational patterns between *Ov*-CCA and non-*Ov*-CCA [9]. Early cancer detection is crucial, and epigenetic therapy can prevent malignant progression. More importantly, within *Ov*-CCA, discernible inter-tumor epigenetic heterogeneity offers the potential for tailoring personalised therapy approaches based on methylation or histone modifications [7].

This epigenetic modification hinders transcription of the methylated DNA, ultimately causing gene silencing in *Ov*-CCA tissues [10]. Although we do not delve into demethylating drugs, palliative chemotherapy, current natural and bioactive compounds, and epigenetic inhibitors involved in ongoing CCA clinical trials (given their absence for *Ov*-CCA), this review holds the potential to propel advancements in infection prognosis, target markers, and therapeutic monitoring for *Ov*-CCA. Hence, this scoping review consolidates recent research findings primarily centered on epigenetic markers and their significance in early detection, prognosis assessment, and potential implicated pathways. These insights offer valuable groundwork for future studies centered on epigenetics in the context of liver fluke-induced CCA.

## 1.1 Rationale

The prognosis for both *Ov*-CCA and non-*Ov*-CCA is poor, given their low resection rate and resistance to traditional radiotherapy and chemotherapy [11]. Subsequent research endeavours in *Ov*-CCA and epigenetics have illuminated a recurring theme. Although abnormal promoter hypermethylation is pervasive in tumor development, the specific genes affected exhibit similarity to the distinctive biology of Opisthorchiasis-induced cancer; however, results may vary according to the gender, age, and grade of an individual patient.

Methylation changes disrupt cell signalling pathways, and it's possible to identify and narrow down affected genes and promoters by focusing on conserved pathways. Until now, there has been no scoping or systematic review of current literature on the role of epigenetic modification in *Ov*-CCA studies. Therefore, we aim to identify and synthesise all available epigenetic evidence through a scoping review. Exploring the broader landscape of this topic will yield more options than concluding limited evidence, aligning with the scoping review's ability to map critical concepts, inquire about types of evidence, and identify gaps across a diverse evidence base. This approach offers an overview of the current state and span of research activity. Furthermore, the results underscore the potential for shared mechanisms in *Ov*-CCA management, emphasising the promise of utilising epigenetic mediators for biomarker discovery, enhancing early detection, and advancing drug development for personalised and combined therapies.

## 1.2. Objectives

- To determine key genes and proteins that exhibit altered epigenetic regulation during opisthorchiasis and *Ov*-CCA development.

- To elucidate the key cellular signalling pathways affected by abnormal epigenetic alterations in *Ov*-CCA development and assess the potential of dysregulated epigenetic markers as diagnostic biomarkers or therapeutic targets.

- To understand the current state of the science regarding epigenetic complications and alterations following *Ov*-CCA and highlight knowledge gaps in the literature to inform future research priorities in this field.

## 2. Methods

### 2.1 Eligibility criteria

This scoping review thoroughly examines studies on individuals diagnosed with *Ov*-CCA and positive tissue sections. The studies exclusively conducted in cell lines, animal studies, review articles, letters, case reports, or without any epigenetic component were excluded from this review. Only articles published in English were selected. The PCC (Population/Concept/Context) framework is employed to structure key concepts, focusing on the crucial involvement of DNA methylation and histone modifications in *Ov*-CCA pathogenesis (S1 Table). The exploration extends to identifying epigenetically regulated genes and pathways linked to opisthorchiasis-induced cancer. This nuanced approach contributes to the overall depth and precision of the scoping review and helps identify key characteristics or factors related to a concept.

### 2.2 Search strategy

The methodology for this study was based on the guidelines outlined by the Preferred Reporting Items for Systematic Reviews and Meta-Analyses for Scoping Reviews (PRISMA-ScR). A comprehensive search was conducted across five electronic databases: PubMed, SCOPUS, Science Direct, and the Cochrane Library. The scoping review presents an overview of a potentially large and diverse body of literature on a broad topic. Therefore, search queries were tailored to encompass the broader context of CCA and its associated epigenetic mechanisms: Epigenetics OR Epigenome* AND Bile duct cancer* OR cholangiocarcinoma. Initially, the search included studies on overall CCA, and subsequently, it was refined to focus on *Ov*-CCA studies that utilised human or human biopsy samples. The last search was done on August 30, 2023.

### 2.3 Data collection

The primary reviewer (AK), with the assistance of MM, conducted the literature search and compilation process. Any disparities in determining the inclusivity of the articles were deliberated upon, and final decisions were reached in consultation with subject matter experts, RKM, SKL, NS, and SS, who possess the requisite knowledge and expertise in tropical medicine and health science. Subsequently, all citations were imported into the desktop reference management application Endnote Reference Manager. Initially, duplicate citations were manually identified and eliminated, along with any duplicates discovered later in the process. Each citation's title, abstract, and keywords were carefully assessed for relevance to the study before being exported into an Excel file along with the corresponding citation records. Any

justifications for omitting citations were duly documented. Complete versions of articles that met the inclusion criteria were procured for further examination. The Excel data extraction form comprehensively recorded the following details: Author (Year), Population, Epigenetic hallmarks, Type of study, Outcome, Method used to measure Outcome, and Summary of the findings.

## 3. Results

In this review, we incorporated all studies investigating epigenetic mechanisms in the context of *O. viverrini*-related CCA. The initial search yielded 1050 records from the four electronic databases. After applying filters specific to each database, duplicates and non-research articles or those not in English were excluded, resulting in 337 remaining records. These were screened based on relevance indicated by their title, abstract, and keywords related to CCA. Additionally, duplicates between databases were removed. Of these, 256 records did not meet the inclusion criteria. Subsequently, 81 records underwent full-text retrieval. The remaining excluded records consist of 12 cell line studies, 49 non-*Ov*-CCA-based studies, 4 non-epigenetic studies, 2 irrelevant studies, 1 report, 1 review, and 1 study on an unrelated topic. Among these, 11 articles were ultimately included in this review, as they utilised human samples and pertained to the causation of *Ov*-CCA (Fig 1). Further confirmation of the use of human-infected *Ov*-CCA tissues was done by sending an e-mail to the corresponding authors of the included studies, as our contextual focus centers on human observational studies conducted in *Ov*-endemic regions within Southeast Asia and tissue samples collected across diverse stages of disease progression.

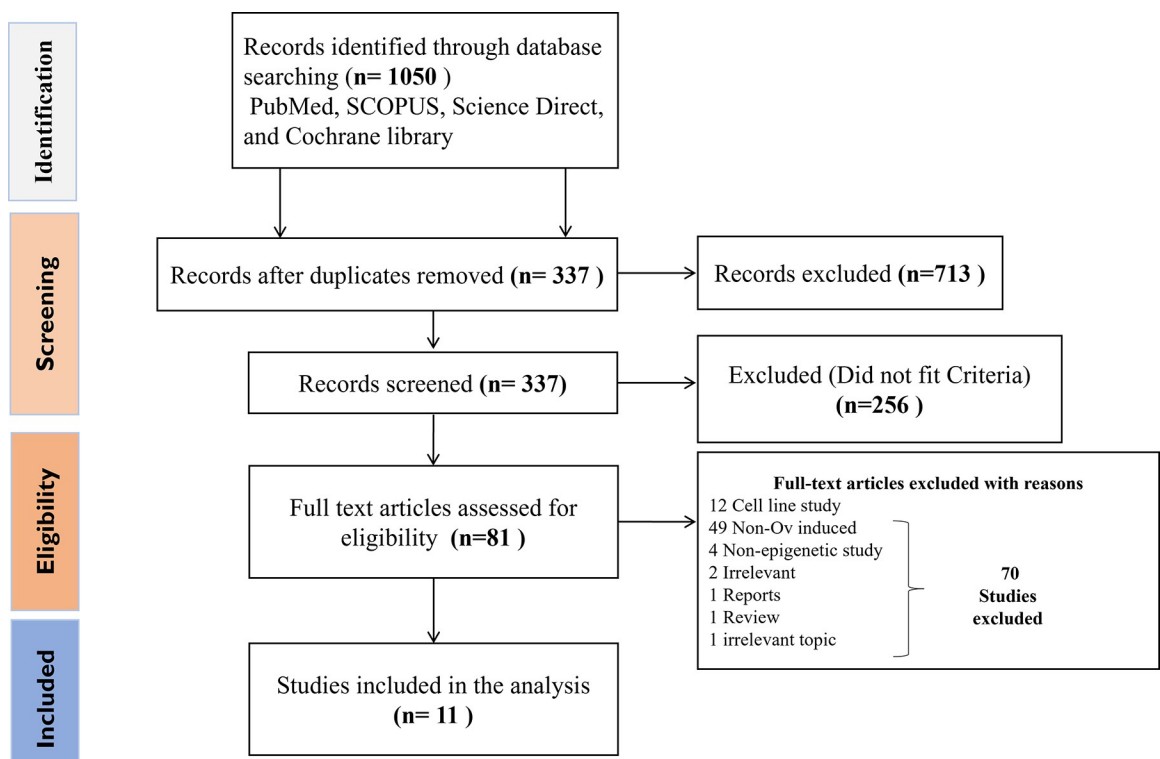

**Fig 1. Flowchart depicting the identification and selection process of studies.**

## 3.1 Sample size and techniques

In the analysis, a total of 860 clinical samples from patients with *Ov*-CCA who tested positive for fluke infection were included from 11 selected studies. It's noteworthy that most of these samples were paraffin-embedded specimens. The core methodologies frequently applied in *Ov*-CCA methylation studies using these clinical samples include PCR-based techniques, bisulfite sequencing, microarray screening, immunohistochemistry (IHC), quantitative PCR (qPCR), and comprehensive computational analysis (S2 Table). None of the studies conducted interventions involving human subjects.

## 3.2 Main findings

In our review, two principal epigenetic mechanisms—DNA methylation and histone modifications—have been identified, with DNA methylation predominantly featured in *Ov*-CCA-based epigenetic research. In our selection of 11 studies focused on *Ov*-CCA and epigenetic aspects, ten studies primarily investigated methylation patterns, with one study based on histone modifications (Table 1).

Additionally, our analysis identified the Wnt/β-catenin pathways as the most prominently affected, both directly and indirectly, in the context of epigenetically dysregulated genes associated with *Ov*-CCA (Table 2). A more comprehensive understanding of recent updates on inhibitors, antagonists, and activators of the Wnt/β-catenin pathway holds promise as a therapeutic approach, given its prominence in *O. viverrini* related epigenetic research.

**3.2.1 DNA Methylation and Histone Modifications: Focusing on Biomarkers and Therapeutic Targets.**

DNA methylation and histone modifications exhibit remarkable potential as biomarkers for early detection, diagnosis, prognosis, and therapeutic monitoring in cancer [12,13]. In *Ov*-CCA tissues, epigenetic modification hinders transcription of the methylated DNA, causing gene silencing and influencing the broader regulatory landscape of cellular processes [10]. DNA methylation markers are detectable in bodily fluids and offer non-invasive approaches to detect cancer onset and recurrence sensitively [14].

Epigenetic studies in *Ov*-CCA have demonstrated abnormalities in DNA methylation with promoter hypermethylation observed in several genes, *hMLH1* [15], Trop2 [16], *RIZ1* [17], *EBF1* [18], *p14A2RF*, *p15INK4B*, *p16INK4A* [19], *PPP4C* (DNA repair), *RUNX3*, *IRF4*, *UCHL1*, *TP53I3* (apoptosis), *CCND2*, *RASSF1* (cell proliferation), *ALDH1A3*, *SLC29A1* (drug metabolism), *HTATIP2* (angiogenesis) [20], *OPCML*, *SFRP1*, *HIC1*, *PTEN*, and *DcR1* [8], *OPCML* [21]. At the same time, *UCHL1* was the only gene found to be hypomethylated in *Ov*-CCA [20].

A detailed analysis of the impact of the aforementioned specific genes on cancer development is presented below, emphasising their potential as markers for disease progression and therapeutic targets. It also highlights their potential as markers in disease progression and therapeutic targets. It further explores their direct or indirect involvement in the Wnt signalling pathways within the *Ov*-CCA pathology and progression context.

## hMLH1 (human MutL homolog 1)

The acquisition of genomic instability stands as a pivotal hallmark of cancer cells, with hypermutable phenotypes known as microsatellite instability (MSI) and mismatch repair (MMR) representing a significant component of this phenomenon [22,23]. Methylation of the hMLH1 gene promoter region can lead to the silencing of hMLH1 expression, resulting in microsatellite instability (MSI) due to dysfunction of the MMR system [24–26]. MMR is crucial for

**Table 1. Summary of the findings from 11 included studies.**

| Author (Year) | Population | Epigenetic hallmarks | Type of study | Outcome | Method used to measure Outcome | Summary of the findings | Country |
|---|---|---|---|---|---|---|---|
| Apinya Jusakul (2017) [63] | Cases: 489 tumors ((133 Fluke-Pos: 132 *O. viverrini*, 1 *C. sinensis*; 39 HBV/HCV-positive; 5 PSC-positive, 312 CCAs of unspecified etiology) | Hypermethylation of CpG islands and CpG island shores | Non-randomized | Cluster 1 and 4 with hypermethylation of CpG islands and CpG island Shore respectively | Multiple omics techniques were used including whole genome sequencing, exome sequencing, targeted sequencing, copy number profiling, DNA methylation profiling and gene expression profiling | The study identified two distinct hypermethylated CCA subgroups: Cluster 1 with CpG island hypermethylation potentially driven by TET1 and EZH2 dysregulation, and Cluster 4 with CpG shore hypermethylation likely due to IDH1/2 mutations. Additionally, Cluster 1 tumors showed a correlation between hypermethylated regions and Signature 1 mutations, indicating disparate somatic evolution between the clusters. Understanding the molecular characteristics of CCA subtypes can lay the groundwork for future research into diagnostic biomarkers and personalized treatment approaches. | Thailand |
| Prasong Khaenam (2009) [17] | 81 CCA tumor tissue samples from patients undergoing surgery, 69 matched non-tumor liver tissue samples | Promoter Methylation (RIZ1) | Non-randomized (retrospective study) | promoter methylation status of RIZ1 and the presence of LOH (loss of heterozygosity) and frameshift mutations in RIZ1 | Methylation-specific PCR, PCR and fragment analysis, and statistical analysis between RIZ1 alterations and patient data. | RIZ1 promoter methylation is common in CCA tumors, and LOH at RIZPro704 is associated with poor patient survival, possibly through the disruption of RIZ1's interaction with estrogen receptor. These alterations could serve as potential biomarkers for prognosis and suggest therapeutic strategies targeting RIZ1 expression | Thailand |

*(Continued)*

**Table 1.** (Continued)

| Author (Year) | Population | Epigenetic hallmarks | Type of study | Outcome | Method used to measure Outcome | Summary of the findings | Country |
|---|---|---|---|---|---|---|---|
| Temduang Limpaiboon (2004) [15] | 65 intrahepatic cholangiocarcinoma tumor samples No control/normal samples | Promoter methylation of hMLH1 gene | Cross-sectional | hMLH1 promoter methylation status Loss of heterozygosity (LOH) and microsatellite instability (MSI) at hMLH1 locus | Methylation specific PCR (MSP) to detect hMLH1 promoter methylation Microsatellite marker analysis to detect LOH and MSI | hMLH1 was frequently methylated in CCA, suggesting epigenetic inactivation is a major mechanism leading to mismatch repair deficiency. hMLH1 methylation status shows potential as a CCA biomarker for diagnostic or prognosis. | Thailand |
| Chaiyachet Nanok (2018) [20] | 54 cholangiocarcinoma (CCA) tissue samples, 19 matched adjacent normal tissue samples | DNA methylation levels of 10 candidate genes involved in various cellular processes PPP4C (DNA repair) RUNX3, IRF4, UCHL1, TP53I3 (apoptosis) CCND2, RASSF1 (cell proliferation) ALDH1A3, SLC29A1 (drug metabolism) HTATIP2 (angiogenesis) | Cross-sectional | DNA methylation levels of the genes HTATIP2 and UCHL1 correlated with overall patient survival | Methylation-sensitive high-resolution melting (MS-HRM) 10 candidate genes involved in DNA repair, apoptosis, cell proliferation, drug metabolism, and angiogenesis) | High HTATIP2 and low UCHL1 methylation correlated with longer survival in CCA patients. Both genes showed higher methylation in CCA tumors than adjacent normal tissues. HTATIP2 and UCHL1 methylation status may be predictive biomarkers for CCA prognosis, with potential therapeutic implications | Thailand |
| Wiphawan Wasenang (2019) [21] | 40 CCA patients, 40 patients with other biliary diseases (control group) | DNA methylation of OPCML, HOXA9, and HOXD9 | Cross-sectional | Methylation levels of OPCML, HOXA9, and HOXD9 in serum cell-free DNA | Methylation-sensitive high-resolution melting (MS-HRM) to quantify methylation levels | Serum methylation levels of OPCML and HOXD9 differ significantly between CCA and controls. OPCML's methylation has high accuracy (AUC 0.850) for CCA diagnosis. Combined OPCML and HOXD9 methylation yields 100% specificity and PPV. Both markers show potential as non-invasive CCA differentiators | Thailand |

(*Continued*)

**Table 1.** (Continued)

| Author (Year) | Population | Epigenetic hallmarks | Type of study | Outcome | Method used to measure Outcome | Summary of the findings | Country |
|---|---|---|---|---|---|---|---|
| Kanlayanee Sawanyawisuth (2016) [16] | 85 cholangiocarcinoma (CCA) tissue samples, 15 matched pairs of normal and CCA tissues for methylation analysis, 6 CCA cell lines | DNA methylation status of the TROP2 gene promoter | Non-randomized | Expression levels and functions of TROP2 in liver fluke associated CCA | Immunohistochemistry (IHC), Bisulfite Genomic Sequencing (BGS), siRNA Knockdown, Cell Proliferation Assay, Migration and Invasion Assays,. Real-time PCR, PCR Array, Western Blotting. | TROP2 exhibited reduced expression in CCA tissues, accompanied by promoter hypermethylation in 60% of cases; its knockdown promoted proliferation and migration of CCA cells, possibly through modulation of genes like MARCKS, EMP1, and FILIP1L, implicating TROP2 in the regulation of proliferation and migration. TROP2 could serve as a potential epigenetic biomarker (diagnostic) or therapeutic target (claimed from cell line based study) in liver fluke-associated cholangiocarcinoma (CCA) | Thailand |
| Ruethairat Sriraksa (2014) [66] | 28 primary intrahepatic CCA samples, 6 matched adjacent normal tissue samples Additional 102 CCA and 24 normal samples for validation | DNA methylation | Non-randomized | Genome-wide DNA methylation profiles Methylation levels of specific genes (e.g. HOX genes) | Illumina Infinium HumanMethylation27 BeadChips, Bisulfite pyrosequencing, Bioinformatic analyses of methylation data | The study found 1610 differentially methylated CpG sites in CCA compared to normal tissues: 809 CpG sites were hypermethylated (603 genes) and 801 were hypomethylated (712 genes). Notably, hypermethylation was enriched at homeobox and PRC2 target genes, including HOXA9 and HOXD9, revealing a stem cell-like pattern. These findings propose HOXA9 genes as potential biomarkers and epigenetic therapies for CCA. | Thailand |

(*Continued*)

**Table 1.** (Continued)

| Author (Year) | Population | Epigenetic hallmarks | Type of study | Outcome | Method used to measure Outcome | Summary of the findings | Country |
|---|---|---|---|---|---|---|---|
| Ruethairat Sriraksa (2011) [8] | 102 primary liver fluke-related cholangiocarcinoma (CCA) samples, 29 adjacent normal tissue samples. Paraffin-embedded sections of CCA samples for immunohistochemistry. | DNA methylation | Non-randomized | Methylation frequency of 26 genes Association with clinicopathological features | MSP, pyrosequencing, COBRA, immunohistochemistry | High methylation frequency of OPCML, SFRP1, HIC1, PTEN, and DcR1 in CCA compared to normal tissue. OPCML methylation is linked to less differentiated CCA, while DcR1 methylation is tied to longer survival. Study shows common promoter hypermethylation in CCA, suggesting OPCML and DcR1 as potential biomarkers for prognosis and treatment response. | Thailand |
| Patcharee Chinnasri (2009) [19] | 94 ICCA patients (63 men, 31 women; median age 54) | DNA methylation of p14ARF, p15INK4b and p16INK4a (tumor suppressor genes (TSGs) located on chromosome 9p21) | Non-randomized, observational study on human samples | LOH and MSI at 9p21-pter Methylation frequency Protein expression by immunohistochemistry | Microdissection, PCR, methylation-specific PCR, immunohistochemistry | Liver fluke–related CCA shows high LOH at 9p21-pter, affecting p14ARF, p15INK4b, and p16INK4a genes. Aberrant methylation leads to their reduced expression. LOH and methylation link to p14ARF loss. p16INK4a loss might predict poor prognosis. DNA methylation of these genes could be diagnostic biomarkers. These genes are potential therapeutic targets for liver fluke related CCA due to their role in disease progression | Thailand |

**Table 1.** (Continued)

| Author (Year) | Population | Epigenetic hallmarks | Type of study | Outcome | Method used to measure Outcome | Summary of the findings | Country |
|---|---|---|---|---|---|---|---|
| Wiphawan Wasenang (2019) [99] | 40 cholangiocarcinoma (CCA) cases Adjacent [38] non-cancerous tissue from the same patients acted as controls | Histone H3 lysine 27 (H3K27) trimethylation by PRC2 complex | Non-randomized | Measurement of EZH2, SUZ12, EED expression | Immunohistochemistry Semi-quantitative scoring (H-score) Kaplan-Meier survival analysis | EZH2 and SUZ12 are highly expressed in CCA compared to adjacent tissue. High EZH2 links to worse survival; combined high EZH2, SUZ12, and EED associated with poorer prognosis, proposing EZH2 as a CCA prognostic biomarker. PRC2 overexpression suggests epigenetic inhibitors for therapy. | Thailand |
| Napat Armartmuntree (2021) [18] | 138 CCA tumor tissues, and 4 normal bile duct tissues | DNA methylation patterns of EBF1 gene promoter region | Non-randomized | EBF1 promoter methylation and expression analyses in patient samples | Methylation-specific PCR on 72 CCA tissues, DNA methylation array data (Illumina 450K) | Targeting EBF1 promoter hypermethylation throC5:J21ugh DNMT inhibition may offer a promising epigenetic therapy for cholangiocarcinoma (CCA) with potential prognostic value as a biomarker, by restoring EBF1 expression and acting as a tumor suppressor to improve outcomes. | Thailand |

precise genome replication during cell division. Deficiencies in this process can cause mutation rates up to 100 times higher than in normal cells [22,27].

The study by Limpaiboon et al. (2005) examined the loss of heterozygosity (LOH) and microsatellite instability (MSI) at the hMLH1 locus and reported hypermethylation in 44.6% of the liver in *Ov*-CCA cases [15]. There had been a research gap in determining the status of *O. viverrini* infection and MMR proteins. A recent study identified a 22.5% prevalence of dMMR in *Ov*-CCA patients [28]. While the study found no statistically significant association between OV IgG status and dMMR protein expression, further research is needed to explore the underlying mechanisms linking dMMR with improved survival and response to therapy. MSI is also an epiphenomenon of dMMR [29], as MSI often results from dMMR. Hence, an in-depth analysis should be considered in *Ov*-CCA tumor types to recognise the beneficial impact of dMMR and MSI in enhancing treatment effectiveness and potential as targets for therapeutic interventions.

## Trophoblast cell surface antigen 2 (Trop2)

The protein Trop2 is produced by the TACSTD2 (Tumor-associated calcium signal transducer) gene [30]. Hypermethylation of the TACSTD2 gene, which encodes Trop2, can silence

**Table 2.  Epigenetic alterations in studied genes/proteins and their impact on critical gene pathways in *Ov*-CCA.**

| Epigenetic Change | Possible major pathways affected |
| --- | --- |
| RIZ1 promoter methylation [17] | -Cell cycle regulation AKT/mTOR [175], IGF-1 (insulin-like growth factor 1) [17,176] and estrogen receptor signalling [46] |
| EBF1 promoter methylation [18] | - B cell development, differentiation, and immune function–HIF1-alpha pathway [177], suppressions of Wnt signalling pathway [18], Potentially PI3K/AKT and other oncogenic signalling [178] |
| hMLH1 methylation [15] | - DNA mismatch repair capability—Microsatellite instability—Potential impact on MAPK, TGFβ, and other tumor-promoting pathways |
| HTATIP2 and UCHL1 methylation [20] | - Angiogenesis signalling through VEGF, HIF1α, etc. [179]—Ubiquitination and protein turnover, association of UCHL1 with β-catenin signalling pathway [180]. UCHL1 could activate the p14ARF-p53 signalling pathway by deubiquitinating p53 and p14ARF [181] |
| OPCML, HOXA9, HOXD9 methylation [21] | OPCML: Wnt signalling inhibitor and functions as tumour suppressor, Wnt/β-catenin and TGF-β-Smad pathways [139], HOX genes: β-catenin/Wnt pathway [182], (AXL)/STAT3 signalling pathways and embryonic development and cell differentiation [68,183,184], [182], HOXD9 [185]. |
| Trop2 methylation [16] | Calcium signalling [186], cell proliferation, and survival—WNT, SRC, AKT, and ERK/MAPK pathways [187] regulating migration extracellular signal-regulated kinase (ERK) [16]. |
| EZH2 overexpression [99] | -Estrogen signalling pathways [133], Aberrant epigenetic repression of tumour suppressors via PRC2 [99], increase the expression of multiple Wnt pathway genes [98] |

its expression, contributing to its downregulation in some cancer types [31]. Trop2 affects cancer cell apoptosis by upregulating Bcl2 and downregulating the expression of the pro-apoptotic protein Bax [32]. The Bax/Bcl-2 expression ratio is a critical determinant in cellular response to apoptotic stimuli. An increased ratio reduces resistance to apoptosis, leading to heightened cell death and a decreased tumor incidence [33].

In *O. viverrini* infection, Bax levels increase at 14 dpi (days post-infection) and decrease at 90 dpi. These fluctuations are associated with the host immune response and the repercussions of recurrent tissue repairs resulting from parasitic infection [34]. The downregulation and internalisation of Bax are associated with metastasis and recurrence [35,36]. Furthermore, during *O. viverrini* infection, NF-κB, generally upregulated [37], and upregulated NF-κB, is known to decrease Bax expression in cancer cell lines [38]. This finding bolsters the evidence showing Trop2 distinct epigenetic silencing in *Ov*-CCA [16]. Reduced Trop2 expression, caused by hypermethylation, could shift the Bax/Bcl-2 ratio, favouring a pro-apoptotic environment. Hence, Trop2, despite its usual overexpression in cancers, is epigenetically methylated and silenced in *Ov*-CCA, making it a unique prognostic marker for *O. viverrini* infection. It also opens up for the treatment enhancing Trop2 antibody-drug conjugates (ADCs) effectiveness through demethylating agents and transcription factor modulation for low Trop2 expression cancers, offering a novel therapeutic strategy [39].

## Retinoblastoma protein-interacting zinc-finger gene 1 (RIZ1)

The RIZ1 gene has been accurately located on chromosome 1q36, a region prone to genetic alterations like deletions, rearrangements, and LOH in many types of human cancers [40–42]. Loss of RIZ1, a methyltransferase targeting lysine nine on histone H3, has been frequently observed in lung, breast, hepatocellular, colon, neuroblastoma, and melanoma, suggesting its potential role as a tumor suppressor in these malignancies [43].

In the case of *Ov*-CCA, Khaenam et al. (2010) showed that RIZ1 hypermethylation may play a potential role in cholangiocarcinogenesis, and the occurrence of RIZPro704 LOH is

linked to poor patient survival in *Ov*-CCA [17]. RIZ1 hypermethylation may dysregulate ERα signalling and drive cancer progression [44, 45]. Compelling research evidence shows the correlation between reduced levels of RIZ1 and dysregulated ERα signalling in cancers [46,47]. Furthermore, higher levels of ER expression (specifically ERα) in cancer cells can lead to increased responsiveness to estrogen and potentially contribute to tumor progression [48,49]. In *Ov*-CCA, male patients exhibited markedly elevated serum estrogen levels compared to controls [50]. This suggests that the combined effects of opisthorchiasis and elevated estrogen levels may exhibit a complex impact on the tumor microenvironment, potentially promoting CCA development and progression.

Using siRNA to downregulate RIZ1 expression showed a correlation between diminished RIZ1 expression and heightened cell proliferation and migration in *Ov*-CCA [51]. The observed inverse epigenetic association with *Ov*-CCA differentiation underscores the potential of RIZ1 expression as a valuable indicator of its stage, aggressiveness, and differentiation.

## EBF1 (Early B-cell Factor 1) in *Ov*-CCA

DNA hypermethylation within the EBF1 promoter region inhibits EBF1 expression and drives the progression of *Ov*-CCA, resulting in severe clinical outcomes [18]. EBF1 is a tumor suppressor in breast cancer [52], leukemia [53], and colorectal cancer [54] and inhibits gastric cancer progression by repressing the telomerase catalytic subunit [55].

Hypermethylation in the EBF1 promoter region correlates with reduced patient survival, repressing its expression and accelerating aggressive CCA advancement [56]. An inverse relationship between EBF1 expression and ZNF423 (Zinc Finger Protein 423) levels in tumor tissues leads to unfavourable prognostic outcomes [57]. In CCA, reduced ZNF423 leads to downregulated MMP9 expression and decreased EMT markers such as N-cadherin and vimentin [58]. Furthermore, ZNF423 is shown to have a complex relationship with EBF1 and can affect gene regulation in B-cell lymphopoiesis, including the downregulation of critical EBF1 targets and potential interference with TGF-β1 signalling [59].

The ability to manipulate EBF1 and ZNF423 levels can influence the disease's course. The intricate relationship between these genes and their impact on the behaviour of the CCA cells strengthens the indication of their potential significance in both diagnosis and targeted treatment.

## TP53 Mutations and ERBB2 Amplification in *Ov*-CCA

TP53 refers to the gene itself, while p53 denotes the protein produced by that gene. Specific missense p53 mutants actively alter the p53 interactome, influencing cellular pathways that promote cancer proliferation, migration, and metastasis [60]. Kiba et al. (1993) found p53 mutation in 35% of Thai CCA cases and suggested that p53 mutation is common in liver fluke related CCA [61]. In *Ov*-CCA tissues, immunohistochemistry analysis revealed an increase in p53 protein levels in 77% of the cases [62].

In a 2017 study, extensive DNA methylation analysis was conducted to investigate the link between multi-omics features and the aetiology of liver diseases [63]. Clusters 1 and 2, linked to liver fluke infection, exhibit common TP53 mutations and amplification of HER2 (ERBB2) in *Ov*-CCA. However, they diverge in CpG island hypermethylation, notably in cluster 1, underscoring the molecular heterogeneity of liver diseases with specific origins.

The HER2 receptor is a 185 kDa transmembrane protein encoded by the HER2, highlighting the importance of targeted therapies. A subset of *Ov*-CCA tissues display protein overexpression of HER2 [64] and exhibit ERBB2 gene amplification. Therefore, examining the impact of HER2 on various *Ov*-CCA subtypes could offer insights and raise the prospect that

infected patients might gain advantages from rapidly evolving anti-HER2-directed therapies. These therapies emphasize personalised approaches, novel agents, and combinations to enhance efficacy while minimising side effects [65].

## HOXD9 and OPCML in Epigenetic therapies and prognosis across cancer types

A study found that hypermethylation was enriched at homeobox and Polycomb Repressive Complex 2 (PCR2) target genes, including homeobox (HOXA9 and HOXD9) [66]. As HOXD9 deregulates histone methylation by PRC2 in cholangiocarcinogenesis, using histone methyltransferase inhibitors to block PCR2 and reactivate silenced HOX genes like HOXD9 could represent a potential epigenetic therapy approach for Ov-CCA. Therefore, understanding the specific methylation patterns and their consequences on HOX genes in Ov-CCA may provide a vital target for therapeutic interventions.

Opioid-binding protein/cell adhesion molecule-like gene (OPCML), an oncogenic suppressor, is frequently silenced by promoter hypermethylation [67]. In 2011, an initial report on OPCML methylation in Ov-CCA revealed a notably high frequency of 72.5%, while no methylation was detected in the adjacent normal tissues [8]. On the contrary, overexpression of OPCML can potentially suppress proliferation and induce apoptosis by deactivating the AXL receptor tyrosine kinase (AXL)/STAT3 signalling pathway in CCA [68]. This demonstrates a robust discriminatory capability of the gene, potentially effective in distinguishing Ov-CCA cancer cases from benign biliary conditions. Additionally, combining OPCML and HOXD9 methylation levels in cfDNA from serum samples provides a less invasive and more effective method for diagnosing and monitoring CCA [21].

## Ubiquitin C-terminal hydrolase L1 (UCHL1)

UCHL1, a member of the UCH class of deubiquitinases (DUBs), has highlighted the importance of promoter region hypermethylation as a mechanism for gene inactivation in cancer [69,70]. UCHL1 is upregulated in Ov-CCA tissue samples and may be linked to hypomethylation in the promoter region and potential induction of UCHL1 expression through cancer-induced DNA repair mechanisms. [20]. High levels of HTATIP2 methylation, also known as CC3 or TIP30, is a gene that encodes a protein involved in various cellular processes. Low levels of UCHL1 methylation were found with extended overall survival in Ov-CCA [20]. Similarly, in gallbladder and breast cancer, UCHL1 exhibits overexpression due to promoter hypomethylation and correlates with metastasis and reduced overall survival [71–73]. These findings align with research linking UCHL1 overexpression to tumor progression, increased size, and invasiveness [74]. Therefore, understanding the molecular alterations associated with UCHL1 hypomethylation-induced upregulation, targeted therapies can be developed to modulate UCHL1 expression, offering a potential avenue for therapeutic intervention in Ov-CCA.

## Loss of Tumor Suppressors p14ARF, p15INK4b, p16INK4a

Separate genes encode p14ARF (p14 Alternate Reading Frame), p15INK4b (p15 Inhibitor of Kinase 4B), and p16INK4a (p16 Inhibitor of Kinase 4A), yet are closely related and functionally connected. CDKN2A encodes p16INK4A and p14ARF, CDKN2B encodes p15INK4B [75]. CDKN2A and CDKN2B have been associated with poorer prognosis in meningioma, acute lymphoblastic leukemia and lung adenocarcinoma [76–78].

Liver fluke CCA cases exhibited notable elevated occurrences of promoter hypermethylation in the p14 (40.2%), p15 (48.9%), and p16 (28.3%) genes [19]. In Ov-CCA, it is characterised by frequent LOH at the chromosomal region 9p21-other, which affects these key tumor

suppressor genes p14ARF, p15INK4b, and p16INK4a [19]. The presence of senescence markers like p14ARF allows the elimination of aberrant aneuploid cells through apoptosis [79]. Still, in *Ov*-CCA, p14ARF is often lost, which may diminish the apoptotic response, allowing the uncontrolled proliferation of abnormal cells and potentially facilitating cancer progression [80].

The methylation of the p16INK4a promoter primarily arises from DNA damage resulting from interference between transcription and replication processes [81]. Therefore, the CDKN2A/B methylation and expression changes show promise as prognostic biomarkers and therapeutic targets related to DNA damage and repair mechanisms in *Ov*-CCA.

## Potential Roles of Hypermethylated DCR1, HIC1, SFRP1 and PTEN in *Ov*-CCA Development

High-frequency methylation of CpG island promoters is a prominent mechanism in intrahepatic CCA (iCCA) [82], influencing various cancer types and contributing to tumor initiation and progression through genome-wide and gene-specific DNA methylation alterations [83]. In *Ov*-CCA, promoter hypermethylation in several CpG islands genes, such as (DcR1, SFRP1, PTEN, H1C1, and DcR1) exhibited a methylation frequency exceeding 28% compared to adjacent normal tissue [8].

Decoy receptor (DcRs) often downregulated through promoter hypermethylation in cancer, overexpression reduces sensitivity to TNF-related apoptosis-inducing ligand (TRAIL) -induced apoptosis and DNA-damaging agents. At the same time, its silencing enhances chemotherapeutic agent-induced apoptosis, indicating its significant role in chemosensitivity regulation [84,85]. In *Ov*-CCA, patients exhibiting methylation at the DcR1 locus displayed a greater survival rate than those without methylation [8,86], indicating the potential use of recombinant TRAIL and TRAIL receptor agonists for *Ov*-CCA treatment. This observation may also be attributed to the role of DcR1 as a decoy, impeding TRAIL binding to death receptors and suppressing the apoptotic process, as observed in tongue carcinoma [87].

The methylation of the promoter region of the secreted frizzled-related protein 1 (SFRP1) holds potential as a diagnostic indicator and therapeutic target, particularly in specific malignancies like ampullary adenocarcinoma and gastric cancer, where its upregulation is linked to poor prognoses [88,89]. In a survey study involving 73 patients diagnosed with primary *Ov*-CCA, the median methylation levels for SFRP1 were 31.5% [90]. The analysis indicated that SFRP1 exhibited a sensitivity of 83.56%, a specificity of 100%, and an overall accuracy of 85.54%. As a secreted protein, detecting SFRP1 methylation in blood or other biofluids could enable non-invasive screening and monitoring of *Ov*-CCA.

The tumor suppressor PTEN loss is common in many cancers, including CCA [91]. PTEN loss in the *Ov*-CCA hamster model shows extensive alterations in DNA methylation and gene expression, ultimately activating the PI3K/AKT/PTEN and Wnt/β-catenin pathways, pivotal in governing cellular growth, proliferation, survival, and metabolism [92]. PTEN helps counteract the activity of PI3K, which is often hyperactivated in opisthorchiasis-induced CCA and promotes cell survival, proliferation, and growth [93]. Hence, hyperactivation of this pathway due to PTEN inactivation may enhance cancer progression during opisthorchiasis.

HIC1 (Hypermethylated in Cancer 1), a tumor suppresser gene, triggered G2/M cell cycle arrest in a glioblastoma cell line model, potentially through induction of full-length p53 and its downstream cell cycle regulators p21 and p27 [94]. In contrast, the truncated p53 isoform Δ133p53 was associated with 5-fluorouracil chemoresistance in *Ov*-CCA models, and its inhibition restored drug sensitivity CCA [95]. Given HIC1's role as a transcriptional repressor that interacts with p53, these findings suggest epigenetic silencing of HIC1 may promote

chemoresistance in *Ov*-CCA by altering p53 regulation and cell cycle control. Therefore, targeting the HIC1-p53 axis therapeutically, through approaches like demethylating agents or Δ133p53 inhibitors, may help enhance response to chemotherapy in *Ov*-CCA patients.

## Histone Modification as a Biomarker and Therapeutic Target in *Ov*-CCA

The "writer" of H3K27me3, Enhancer of zest homolog 2 (EZH2), has been known to be involved in tumor progression and is the catalytic subunit of Polycomb repressive complex 2 (PRC2), which is a highly conserved histone methyltransferase that targets lysine-27 of histone H3 [96,97]. Mice exhibiting liver specific EZH2 knockout demonstrated reduced CCA development, while also, in a xenograft model, EZH2 knockdown markedly slowed the progression of CCA [98]. Similarly, in *Ov*-CCA tissue samples, high EZH2 expression is associated with poorer survival, and combined high EZH2, SUZ12, and EED correlated with a worse prognosis [99].

The unique upregulation of H3K27me3 by the PRC2 complex in *Ov*-CCA, notably involving EZH2 and SUZ12 compared to other cancers, signifies a specific epigenetic mechanism contributing to its aggressiveness, potentially making histone-modifying enzymes like EZH2 viable targets for treatment and prognosis evaluation [99]. Dysregulation of the PRC2–H3K27me3 axis is linked to different diseases, including several cancers [96,100] and inflammatory processes [101]. Therefore, SUZ12 and EED are other essential components of the PRC2 complex. SUZ12 stabilises PRC2 and is necessary for its methyltransferase activity [102], and EED interacts with EZH2 and helps guide PRC2 [103] to its target sites on chromatin [104]. Recently, compounds that disrupt the interaction between EZH2 and EED have been developed, leading to the destabilisation and degradation of PRC2 proteins, which could provide a new avenue for cancer therapy [104,105]. These compounds represent a range of strategies for targeting components of the PRC2 complex, offering potential therapeutic avenues for various cancers, including CCA and *Ov*-CCA.

In summary, both DNA methylation and histone epigenetic alterations appear as crucial events in *Ov*-CCA (Fig 2). However, it's important to note that regulating the genes and proteins is complex and context-dependent. The specific pathways and mechanisms involved in their upregulation or downregulation may vary in different cell types and under other conditions. Understanding these regulatory mechanisms is essential for developing targeted therapies and diagnostic approaches for diseases involving their dysregulation.

**3.2.2 Gene Set Enrichment Analysis of Epigenes in *Ov*-CCA.** The genes identified in this review, implicated in *Ov*-CCA pathology, underwent Enrichment analysis using Enrichr (https://maayanlab.cloud/Enrichr/) against the Wiki Pathway Collection library [106], an open-access platform supported by a collaborative scientific community. In total 23 genes (hMLH1, RIZ1, Trop2, ERBB2, EBF1, OPCML, HOXA9, HOXD9, HTATIP2, UCHL1, p14ARF, p15INK4b, p16INK4a, EZH2, SUZ12, H3K27, PTEN, H1C1, PRC2, DKK1, DCR1, HIC1, SFRP1) identified in *Ov*-CCA epigenetic alterations were subjected to gene set enrichment analysis (Fig 3A). The top five predicted pathways based on highest combined score were selected and shown in a bar chart (Fig 3B).

**3.2.3 Identifying Hub Genes and Pathway Dysregulation in *Ov*-CCA.** To be more species specific, the STRING database [107] was used to predict protein-protein interactions involving these genes (Fig 3A), which includes both physical interactions and functional associations. These genes were analyzed with the most deregulated pathways related to *Ov*-CCA. These pathways were compiled from literature reviews and underwent manual screening of titles and abstracts, using specific keywords. The most deregulated pathways based on *Ov*-CCA studies were PI3K (Phosphoinositide 3-kinase), AKT (protein kinase B), mTOR

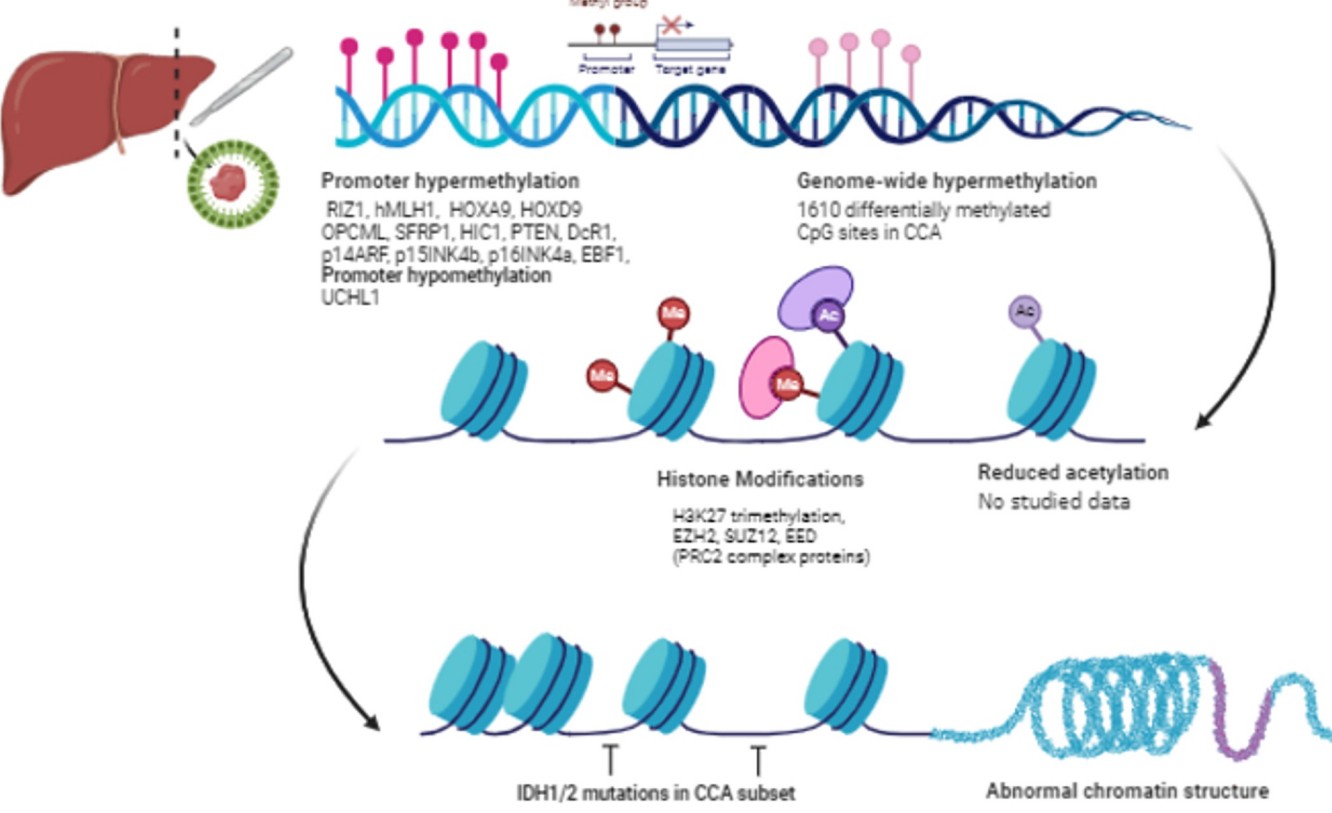

**Fig 2. Genes and Proteins involved in epigenetic regulation of *Ov*-CCA (Created with BioRender.com).**

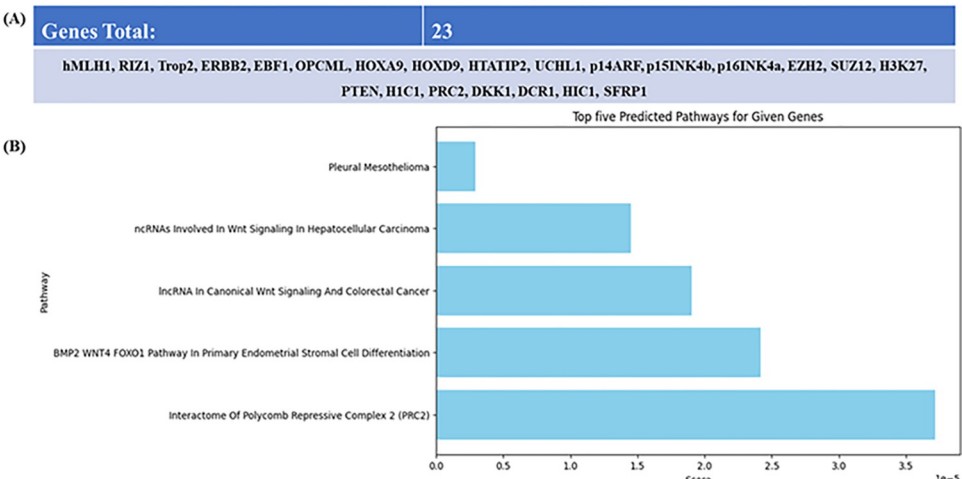

**Fig 3.** (A) *Ov*-CCA-associated genes (B) Top five predicted pathways for genes involved in *Ov*-CCA. This analysis revealed pathways exhibiting the greatest overlap with the queried gene set. Notably, Wnt signalling appeared twice in the chart with the highest combined score. A higher combined score indicated lower p-values (greater statistical significance) and increased gene overlap, reinforcing the enrichment confidence. These research findings also merit researchers investigating genes and pathways in pleural mesothelioma.

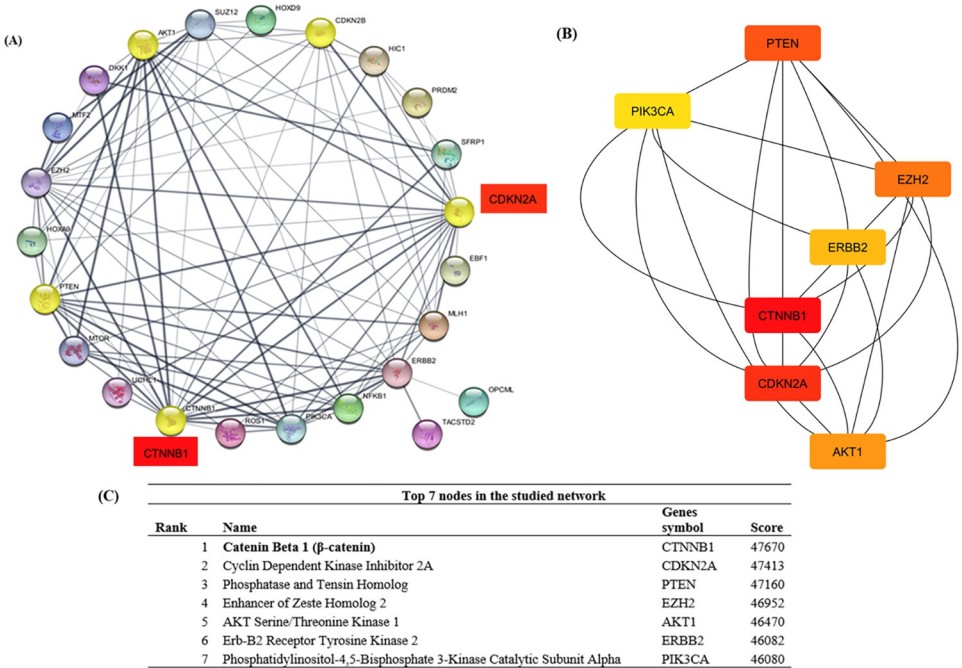

**Fig 4.** (A) STRING interaction of genes and pathways involved in *Ov*-CCA. (B)Top 7 genes showed in Hierichial layout. (C) Ranked gene names and string interaction scores.

(mammalian target of rapamycin), PTEN (phosphatase and tensin homolog), CTNNB1 (Catenin Beta-1), NAKB1 (Nuclear Factor Kappa B Subunit 1), MAPK (Mitogen-Activated Protein Kinase), and RTK (Receptor Tyrosine Kinase).

In the STRING database, hub genes exhibit extensive connectivity within a network, underscoring their pivotal role and significant impact on biological processes through interactions with multiple partners. β-catenin (CTNNB1) demonstrated elevated centrality values relative to other ranked genes in the network, suggesting its potential as a target for pathway-specific therapies or drug interventions in *Ov*-CCA (Fig 4B). CTNNB1 showed the highest score at 47670 (Fig 4C). A higher score suggests the gene likely acts as a central hub in biological networks, facilitating interactions across various pathways and processes.

### 3.2.4 Wnt/β-catenin signalling pathway emerges as a common signalling pathway in *Ov*-CCA.

The Wnt pathway utilises canonical (Wnt/β-catenin) and noncanonical (Wnt/Ca2+ and planar cell polarity) routes. The canonical pathway involves β-catenin activation of genes through TCF/LEF. and noncanonical pathways regulate cell polarity and migration independently of β-catenin-TCF/LEF, forming a distinct network from the canonical pathway's control over cell proliferation. [108].

Wnt ligands play a pivotal role by interacting with cell membrane-bound receptors and co-receptors, influencing critical physiological and pathophysiological functions across various organs and cell types. These functions include organ development, cancer formation, and fibrosis [109]. In molecular oncology research, comprehending the underlying mechanisms of the anti-tumor effects and pathways involved in overall CCA progression is pivotal for tailoring targeted therapies, ensuring treatment efficacy, and reducing adverse effects [110,111]. Our analysis showed that Wnt/β signalling predominantly emerges in *Ov*-CCA (Table 2).

Most of the discussed genes/proteins appear to function by suppressing Wnt/β-catenin signalling in *Ov*-CCA. In *Ov*-CCA research, although hMLH1 isn't directly linked to the Wnt signalling pathway, the hypermethylation of the DNA mismatch repair (MMR) gene MLH1's promoter and microsatellite instability (MSI)-related tumors have been correlated with WNT signalling activation, likely through frameshift insertion/deletion mutations [112]. This activation can result in reduced tumor-infiltrating lymphocytes (TILs) and resistance to immune checkpoint inhibitor (ICI) therapy [113,114], suggesting a less favourable environment for the immune system to mount an effective anti-cancer response. The loss of hMLH1 in *Ov*-CCA is mainly associated with DNA repair [115], and deviant methylation patterns within DNA damage repair genes hold the potential to function as predictive, prognostic, and chemosensitive markers in the context of human cancer [116].

In human tissue samples of *Ov*-CCA, RIZ1 gene methylation and epigenetic alteration were associated with poorer survival [17,51]. RIZ1 suppresses Wnt signalling via β-catenin in breast cancer and parathyroid tumors [117,118], implying a conserved negative regulatory role of RIZ1 in the β-catenin pathway across these malignancies. Therefore, targeting Wnt signalling and restoring RIZ1 function shows potential in overall CCA treatment. However, contextual factors like age, gender, and organ may influence RIZ1's function, as in estrogen-related cancer, as it exhibits a dual role, acting as both a promoter and inhibitor of tumor growth, highlighting its complex and context-dependent nature [119].

In *Ov*-CCA, Trop2 is frequently hypermethylated, causing growth and metastatic advantages to the biliary cancer cells [16], which sets them apart from other cancer types. Trop2 co-localizes with β-catenin in the nucleus, upregulating cyclin D1 and c-myc, fostering nuclear oncogene transcription and cell proliferation [120]. The Wnt-β-catenin signalling pathway can modulate the activation of β-catenin, a pivotal element in Wnt signalling, thereby indirectly influencing the Wnt signalling cascade [121]. Therefore, the unique hypermethylation of Trop2 in *Ov*-CCA cells implies that it does not play its usual role in interacting with the Wnt/β-catenin pathway to affect cancer progression but rather a suppressive effect fostering tumorigenesis in *O. viverrini* infection.

EBF1 is frequently hypermethylated and silenced epigenetically in its promoter region in *Ov*-CCA tissues [18]. EBF1 is linked to regulating genes associated with the immune response and cytokine production, including IL-6 [122]. Furthermore, the IL-6/STAT3 signalling pathway is implicated in the hypermethylation of the EBF1 promoter, demonstrating that IL-6 potentially plays a role in this epigenetic modification [123]. IL-6 also can interact with Wnt signalling pathways, aiding in repair and regeneration [124]. Both IL-1β and IL-6 are widely produced in *O. viverrini* infection [125,126], and these interleukins have significantly reduced Dickkopf-1 (DKK1) production using Wnt signalling pathway, potentially causing an abnormal Wnt pathway [127]. Additionally, in bioinformatics analyses of gene expression data sets in intrahepatic (iCCA) and extrahepatic (eCCA) cholangiocarcinomas, DKK1 was identified alongside MMP7 as a distinguishing marker between these cancer types [128].

Aberrant Wnt activation drives abnormal cell growth and neoplastic transformation, while concurrent CDKN2A/B deficiency synergistically promotes tumorigenesis via Wnt signalling pathway activation [129]. This combined effect substantially increases the risk of cancer, highlighting the pivotal roles of both elements in suppressing tumors. In *Ov*-CCA, aberrant p14ARF methylation, resulting from DNA damage, reduces its expression, impacting genomic stability. CDKN2A and TP53 are crucial, with p14ARF from CDKN2A enhancing p53's tumor-suppressing functions [130]. Additionally, the loss of the CDKN2A/B locus, which encodes essential cell cycle inhibitors like p16INK4A and p15INK4B, is common in *Ov*-CCA. This loss may cause the co-deletion of the tumor suppressor gene methylthioadenosine phosphorylase (MTAP), resulting in uncontrolled cell proliferation [131], as MTAP is located on

the 9p21 chromosome, where CDKN2A/B is also located.CDKN2A/B and MTAP loss, p16INK4A methylation, senescence marker downregulation, and p53 alteration reflect fundamental genomic and epigenetic changes in *Ov*-CCA pathogenesis.

EZH2 silences Wnt pathway antagonists, activating Wnt/β-catenin signalling in hepatocellular carcinomas, contributing to their proliferation and suggesting a potential therapeutic target [132]. EZH2 transactivates genes commonly targeted by estrogen and Wnt signalling pathways [133]. In the context of anaplastic thyroid carcinoma (ATC), the inhibition of EZH2 has been demonstrated to suppress Wnt/β-catenin signalling, thereby contributing to its oncogenic role, with targeting EZH2 resulting in reduced β-catenin activity and consequential effects on proliferation and invasion [134]. However, in the context of *Ov*-CCA, EZH2 is overexpressed, potentially disrupting balanced Wnt/β-catenin signalling and resulting in heightened β-catenin downstream signalling. Similarly, in cervical cancer, the overexpression of EZH2 promotes tumor progression by enhancing cell proliferation and tumor formation through activation of the Wnt/β-catenin pathway, achieved via epigenetic silencing of GSK-3β and TP53 [135]. In chemoresistant Head and Neck Squamous Cell Carcinoma (HNSCC) cells, both the Wnt/β-catenin pathway and EZH2 are upregulated, indicating a relationship between higher EZH2 expression and Wnt/β-catenin pathway activation, contributing to chemoresistance and cancer stem cell accumulation [136].

SFRP1, in *O. viverrini* infection, is known to serve as a Wnt inhibitor and is overexpressed in EZH2-deficient tissues [137]. Wnt proteins initiate intracellular signalling pathways by binding to frizzled (FZ) receptors and coreceptors, thus revealing their cellular effects [138], and notably, in the case of *Ov*-CCA, it has been observed that SFRP1 is downregulated from hypermethylation, supporting the finding of EZH2 overexpression leading to decreased SFRP1. The upregulation of EZH2 and downregulation of SFRP1 in the context of *O. viverrini* infection may play a crucial role in aberrant expression of the Wnt/β-catenin pathway, and it may also be associated with the activation of Wnt/β-catenin signalling and the consequential proliferation of CCA cells. Regulating this mechanism can potentially control cell proliferation, migration, and invasion in opisthorchiasis while maintaining an effective immune response that supports the survival of the host-parasite relationship. These findings underscore the epigenetic significance of SFRP1 and EZH2 unique to *Ov*-CCA and suggest potential avenues for therapeutic intervention.

In *Ov*-CCA, the highest methylation frequency of OPCML was observed (72.5%), indicating its potential use as an epigenetic biomarker for CCA prognosis and diagnosis. In the context of esophageal cancer, OPCML's downregulation in Grade 3 tumors may implicate its regulatory role in the Wnt/β-catenin signalling pathway, potentially contributing to the development and progression of higher-grade tumors [139]. This relationship remains unexplored in *Ov*-CCA tissue samples and cases. OPCML, also a stress- and p53-responsive gene, when hypermethylated, diminishes the response; introducing OPCML in carcinoma cells lacking it results in substantial growth inhibition, highlighting its potential as a tumor suppressor [140]. A Mutant p53 (mutp53) protein can interact with components of the Wnt pathway, such as β-catenin, to enhance their activity.

Generally, TP53 mutations activate the Wnt pathway via mutp53 interaction [141], fostering poorly differentiated and invasive tumors in cancer. Without TP53's normal tumor suppressor function, mutp53 can influence other pathways, including Wnt, that drive cancer progression [142]. Missense TP53 mutations directly affect the Wnt pathway, and in *O. viverrini* infection, exon 6 TP53 mutations may contribute to cholangiocarcinogenesis [143]. Hence, elevating components of the Wnt pathway through the attenuation of p53 signalling has been demonstrated to be a critical factor in promoting the formation of metastases [144]. Crosstalk between the p53 and Wnt pathways involves direct interactions, where p53 can

regulate the stability and subcellular location of β-catenin. Additionally, these pathways affect each other's protein stability and gene expression, influencing essential cancer-related genes [142].

Cancer research actively investigates potential treatment drugs to modulate the Wnt/β-catenin signalling pathway, a key player in cancer progression. These drugs can either inhibit or activate Wnt signalling, depending on the specific disease type, stage, and lesion characteristics, as highlighted by recent studies [145,146]. Further exploration is needed to determine whether targeting epigenetically activated genes and proteins could potentially suppress *Ov*-CCA tumors by regulating the Wnt pathway and whether the epigenetic status of these genes and proteins could serve as a useful biomarker in this context.

## 4. Wnt/β-catenin as a therapeutic target for *Ov*-CCA

Extensive research has uncovered genetic alterations in cancer-related signalling pathways, leading to the development of targeted therapies [147]. Using String-DB [148], network analysis revealed a notable interaction pattern among the 23 dysregulated epigenes, notably with β-catenin and Cyclin Dependent Kinase Inhibitor 2A (CDKN2A) ranking second highest in the analysis (Fig 4).

Wnt/β-catenin signalling, known for directly or indirectly suppressing various Cyclin-Dependent Kinase Inhibitor proteins (CKIs, or CDKNs). For example, p16, encoded by p16INK4A (CDKN2A), is an inhibitor of CDK4, and the expressions of cyclin D1 and CDK4 are up-regulated in opisthorchiasis-associated CCA [149]. There is a possibility that the expression of β-catenin may prevent senescence by suppressing p16INK4A expression [150]. Therefore, the downregulation of p16INK4A may allow cancer cells to bypass senescence and continue proliferating, contributing to tumor progression. Furthermore, the Wnt target, MYC, inhibits the transcription of several other CKIs, including CDKN1A (p21), CDKN1B (p27), and CDKN2B (p15) [151], all of which demonstrate hypermethylation *Ov*-CCA and CCA.

Various strategies to inhibit this pathway are being explored in preclinical and clinical studies, including blocking porcupine enzymes, silencing DKK1, targeting microRNAs, and regulating the PI3K/AKT/PTEN/GSK-3β, retinoic acid receptor (RAR), protein kinase A regulatory subunit 1 alpha (PRKAR1A/PKAI), liver kinase B1 (LKB1) and CXCR4 axes that modulate β-catenin activity and its downstream target genes [152,153]. Therefore, investigating whether the dysregulation mentioned above is a shared feature among different causative agents of *Ov*-CCA would provide valuable insights for the potential use of Wnt/β-catenin targeted therapies in the overall management of CCA.

Dickkopf-1 (DKK1), a soluble antagonist of the Wnt/β-catenin signalling pathway, is elevated in the serum of patients with various cancers and animal models of chronic inflammatory diseases [154]. The bidirectional relationships between DKK1 and inflammation, influenced by systemic inflammatory signals and local tumor inflammation, highlight its intricate involvement in disease pathways [155]. Considering its association with immunosuppressive phenotypes in iCCA [156], DKK1 emerges as a potential therapeutic target for iCCA [157]. In *O. viverrini* infection, IL-6 and IL-1β are upregulated [158], and both of these interleukins may significantly dampen Wnt antagonist Dickkopf-1 (DKK1) [127,159]. The activation of Wnt signalling via downregulation of DKK1 may impact downstream pathways, such as NF-κB, which govern inflammatory responses [160] during *O. viverrini* infection. Therefore, exploring DKK1's role in *Ov*-CCA and its impact on NF-κB and Wnt/β-catenin crosstalk is crucial for identifying potential anti-cancer therapy targets.

The suppression of PRKAR1A expression resulted in growth inhibition and apoptosis of *Ov*-CCA, while its overexpression was correlated with elevated levels of ECPKA autoantibodies [161]. ECPKA shows potential as a cancer biomarker, with the prospect of being utilised to differentiate between malignant tumors and benign conditions [162]. Furthermore, silencing PRKAR1A reduces multiple signalling pathways and can affect various intracellular signalling pathways, including MAPKs, PI3K/Akt, JAK/STAT, and Wnt/β-catenin, influencing CCA cell behaviour and growth [163]. These discoveries highlight the potential of targeting the PKA pathway to regulate Wnt/β-catenin in *Ov*-CCA therapy, alone or in combination with other anticancer drugs.

CXCR4, a chemokine receptor, is involved in downstream signalling triggered by the interaction of CD63 with N-glycans found on tetraspanin (TSPs) released by *O. viverrini* [164]. Elevated CXCR4 expression has been linked to iCCA progression and metastases [165], and CXCR4 knockdown has been correlated with the suppression of Wnt target genes and the inhibition of malignant biliary tract tumor progression [165,166]. A better understanding of how *O. viverrini* infection influences CXCR4 expression and associated signalling cascades may provide key insights into and strategies to suppress CXCR4-driven Wnt activation and disease progression in *Ov*-CCA patients.

Extensive research has uncovered genetic alterations in cancer-related signalling pathways, leading to the development of targeted therapies [147]. Dysfunctional Wnt/β-catenin signalling is a crucial pathway in numerous cancers and diseases, impacting cell proliferation, invasion, and stemness and leading to enduring epigenetic alterations in gene expression [108,167]. Our review study identified all epigenetic-based studies on opisthorchiasis-induced CCA, the types of epigenetic modifications involved, and their potential as biomarkers and therapeutic applications (Table 2). Through this analysis, we observed that the Wnt/β-catenin signal transduction pathway is frequently dysregulated in *Ov*-CCA, primarily due to promoter methylation. Given that the Wnt/β-catenin pathway has emerged as a central player in CCA, its alteration is associated with worse outcomes in specific CCA subtypes [152]. Hence, exploring the complex Wnt signalling pathway offers promising and innovative therapeutic opportunities for treating *Ov*-CCA, as well as non-*Ov*-CCA. Although clinical trials testing Wnt pathway inhibitors have been undertaken, they have not yet received approval. The primary obstacle lies in the remarkable evolutionary conservation of the canonical WNT/β-catenin signalling pathway, a pivotal regulator of tissue development and homeostasis [168]. Nevertheless, some trials combining Wnt inhibition with chemotherapeutic drugs have shown promise [169].

## 5. Discussion and conclusion

In the post-genomic era, the epigenetic foundation of cancer development has brought about a significant transformation in cancer genetics. This breakthrough has introduced new avenues for therapeutic interventions in cancer treatment [170]. This scoping review investigates the genes associated with epigenetic changes in the progression of liver fluke-induced cancer, examining their interplay with Wnt- β-catenin signalling in *Ov*-CCA. The study provides valuable insights into potential biomarkers and therapeutic targets related to epigenetic studies in *O. viverrini* infection. The Wnt/β-catenin signalling pathway offers a promising target for opisthorchiasis-induced cancer therapy, given its central role in cancer progression and its potential as a focus for inhibitors. The analysis also elucidates strategies leveraging epigenetic markers for therapeutic interventions. While its involvement in carcinogenesis is well-characterized, aberrant Wnt signalling is also observed in various other cancer types.

Wnt/β-catenin signalling is central to cancer research for its key role in cancer development and progression. In this paper, we systematically examined studies investigating the epigenetic

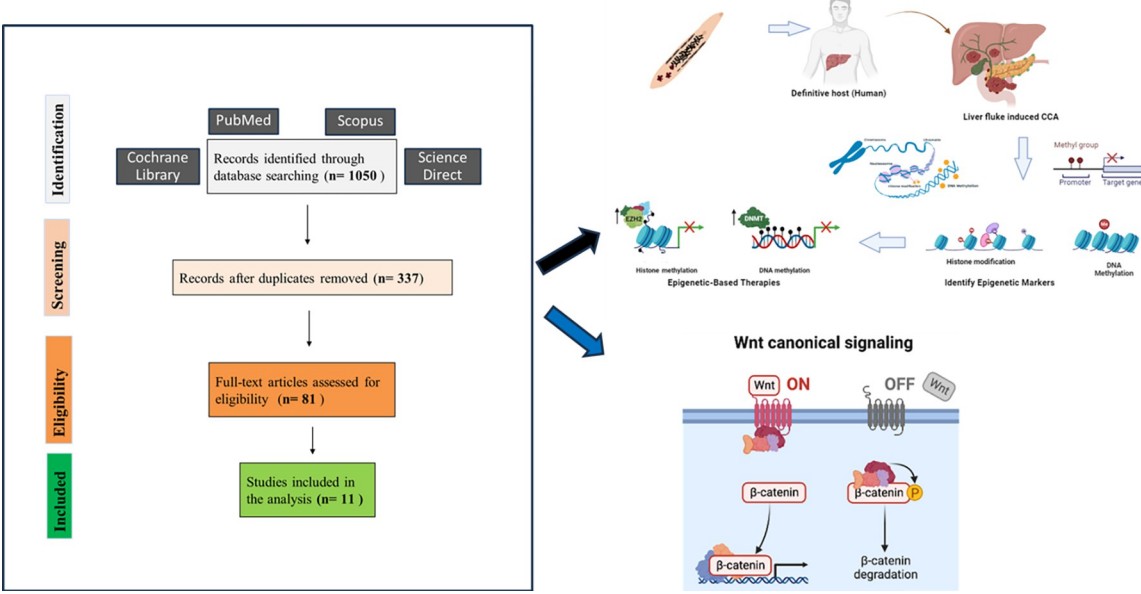

**Fig 5. This review is the first to scope, collate, and catalogue research investigating epigenetic markers for diagnosis and therapeutic potential in *Ov*-CCA.** It identifies several potentially valuable epigenetic markers and potential drug targets, encouraging further exploration. Additionally, it sheds light on shared signalling pathways that could help design future studies focused on epigenetics in this field. The findings of this research present promising avenues for developing epigenetic-based therapeutic strategies and differential biomarkers in the treatment of *Ov*-CCA. (Created with BioRender.com).

mechanisms underlying *Ov*-CCA (Fig 5). All clinical samples in our study were obtained from patients diagnosed with *Ov*-CCA and were analyzed across selected studies. The predominant epigenetic alterations involved DNA hypermethylation of promoter CpG islands, notably impacting the expression of tumor suppressor genes. Numerous genes displayed hypermethylation in *Ov*-CCA tissues compared to normal tissues, including hMLH1, RIZ1, Trop2, EBF1, p14ARF, p15INK4b, p16INK4a, OPCML, PTEN, and SFRP1. Gene set enrichment analysis revealed the Wnt/β-catenin pathway as the most significantly enriched pathway associated with these differentially methylated genes. Moreover, network analysis identified β-catenin as a central hub gene interacting with multiple epigenetically altered genes. Collectively, these findings suggest dysfunctional Wnt/β-catenin signalling as a common pathological mechanism in *Ov*-CCA, primarily driven by aberrant promoter hypermethylation. Since the Wnt pathway regulates critical processes like proliferation, pathogenesis, invasion, and stemness, targeting this pathway may offer therapeutic benefits for *Ov*-CCA. However, medications targeting Wnt/β-catenin signalling in cancers are either not yet in clinical trials or are in early-phase trials with unsatisfactory outcomes [171]. In conclusion, this scoping review enhances understanding of the epigenetic landscape in *Ov*-CCA and provides insights towards developing novel targeted and precision therapies for this disease.

In recent years, there has been an increase in the incidence of CCA, yet its prognosis remains challenging [172]. This is primarily due to its limited surgical resection success rate and its resistance to traditional radiotherapy and chemotherapy methods, which pose significant challenges for effective treatment [173]. Unlike genetic mutations, epigenetic alterations are reversible and dynamically influenced by the tumour microenvironment, making them attractive candidates for minimally invasive testing and disease progression tracking. Hence, understanding the underlying mechanisms driving CCA pathogenesis induced during *O. viverrini* infection, the factors governing its growth, signalling pathways, and leveraging

epigenetics offer a promising avenue for personalised, targeted therapies and enhanced diagnostic and prognostic capacities in disease management. Ultimately, this study intends to offer guidance for future research by identifying key considerations and insights essential for developing more targeted therapies for Wnt/β-catenin signalling in cancer.

## 6. Limitations

Extensive research has elucidated that canonical and non-canonical Wnt signalling impacts cancer by interacting with the microenvironment and immune system, with non-canonical signalling regulating cell motility in both development and metastasis, while their balance varies across different tissues and tumours [174]. Moreover, variability in laboratory procedures across different studies could have contributed to variations in test results, and it's important to note that the analysis was primarily performed on samples from the primary tumour, potentially introducing intra-tumour heterogeneity, as the papers did not specify whether patients were being treated for their metastases. Furthermore, this review did not examine small noncoding RNAs like microRNAs and long noncoding RNAs (lncRNAs). These can serve as multifactorial epigenetic gene regulators by modifying transcription and post-transcriptional regulation and signalling pathways. The absence of research addressing these aspects in *Ov*-CCA and epigenetic markers is a notable limitation within both the scope of *Ov*-CCA studies and this specific scoping review. Further future research should focus on distinguishing *Ov*-CCA-specific methylation changes from those influenced by non-*Ov*-CCA factors using comprehensive techniques like methylated DNA immunoprecipitation microarray (MeDIP-chip) genome-wide analysis. This can help identify definitive diagnostic epigenetic markers for CCA and advance our understanding of this disease. The absence of precise numerical data while studying epigenetic alteration reported in *O. viverrini* lacks in-depth statistical analysis, including sensitivity, specificity, AUC, OR, RR, and survival rates, within the included study text precludes a comprehensive assessment of the biomarker efficacy of epigenetic markers or their therapeutic prospects.

## Supporting information

**S1 Table. The PCC (Population/Concept/Context) framework employed to structure key concepts, focusing on the crucial involvement of DNA methylation and histone modifications in *Ov*-CCA pathogenesis.**
(DOCX)

**S2 Table. The core methodologies frequently applied in *Ov*-CCA methylation studies.**
(DOCX)

## Author Contributions

**Conceptualization:** Alok Kafle, Sutas Suttiprapa.

**Data curation:** Alok Kafle, Mubarak Muhammad, Jan Clyden B. Tenorio.

**Formal analysis:** Alok Kafle, Mubarak Muhammad, Jan Clyden B. Tenorio, Roshan Kumar Mahato.

**Funding acquisition:** Sutas Suttiprapa, Norhidayu Sahimin, Shih Keng Loong.

**Investigation:** Alok Kafle.

**Methodology:** Alok Kafle, Mubarak Muhammad, Jan Clyden B. Tenorio, Roshan Kumar Mahato.

**Validation:** Sutas Suttiprapa, Norhidayu Sahimin, Shih Keng Loong.

**Visualization:** Alok Kafle, Sutas Suttiprapa.

**Writing – original draft:** Alok Kafle, Sutas Suttiprapa.

**Writing – review & editing:** Alok Kafle, Sutas Suttiprapa, Roshan Kumar Mahato, Norhidayu Sahimin, Shih Keng Loong.

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
