## [Decision Letter · Decision Letter 0]

3 Jun 2024

Dear Dr. Suttiprapa,

Thank you very much for submitting your manuscript "Epigenetic Biomarkers and the Wnt/β-Catenin Pathway in Opisthorchis viverrini-associated Cholangiocarcinoma: A Scoping Review on Therapeutic Opportunities" for consideration at PLOS Neglected Tropical Diseases. As with all papers reviewed by the journal, your manuscript was reviewed by members of the editorial board and by several independent reviewers. In light of the reviews (below this email), we would like to invite the resubmission of a significantly-revised version that takes into account the reviewers' comments. 

We cannot make any decision about publication until we have seen the revised manuscript and your response to the reviewers' comments. Your revised manuscript is also likely to be sent to reviewers for further evaluation.

Sincerely,

Aysegul Taylan Ozkan, M.D., Ph.D.,

Academic Editor

Richard Bradbury

Section Editor

Reviewer's Responses to Questions

**Key Review Criteria Required for Acceptance?**

**Methods**

-Are the objectives of the study clearly articulated with a clear testable hypothesis stated?

-Is the study design appropriate to address the stated objectives?

-Is the population clearly described and appropriate for the hypothesis being tested?

-Is the sample size sufficient to ensure adequate power to address the hypothesis being tested?

-Were correct statistical analysis used to support conclusions?

-Are there concerns about ethical or regulatory requirements being met?

Reviewer #1: (No Response)

Reviewer #2: This review meet the above method conditions.

Reviewer #3: Yes

Yes

Yes

Yes

Yes

Yes

**Results**

-Does the analysis presented match the analysis plan?

-Are the results clearly and completely presented?

-Are the figures (Tables, Images) of sufficient quality for clarity?

Reviewer #1: (No Response)

Reviewer #2: Yes.

Reviewer #3: Yes

Yes

Yes

**Conclusions**

-Are the conclusions supported by the data presented?

-Are the limitations of analysis clearly described?

-Do the authors discuss how these data can be helpful to advance our understanding of the topic under study?

-Is public health relevance addressed?

Reviewer #1: (No Response)

Reviewer #2: Yes

Reviewer #3: Yes

Yes

Yes

Yes

**Editorial and Data Presentation Modifications?**

Reviewer #1: (No Response)

Reviewer #2: accept

Reviewer #3: Review comment

This review article is well organized and is appropriate to the objective.

However, there are some points to be questioned, as follows:

1. In Section “3. Results”, page 9.

“The initial search yielded 495 records from the four electronic databases.”

 In Figure 1. Flowchart depicting the identification and selection process of studies, page 10.

 “Records identified through database search (n=1050)” 

The number “495 records” and “n = 1050”, which number is an initial searching record? 

2. In Section “3.1 Sample Size and Techniques”, page 10

“immunohistochemistry (IHC), quantitative PCR (qPCR), and comprehensive computational analysis (supplementary 2)”

Is supplementary 2 refer to Supplementary Table 2?

3. In “TP53 Mutations and EBB2 Amplification in Ov-CCA”, page 15

Is EBB2 the same gene as (ERBB2) described in the text?

4. In Section “5. Conclusion and perspectives:”, page 31

The authors have mentioned that “Unlike genetic mutations, epigenetic alterations are reversible and dynamically influenced by the tumour microenvironment, making them attractive candidates for minimally invasive testing and disease progression tracking.”

How does tumor microenvironment influence epigenetic alterations? Or what evidence that support this tumor microenvironment related epigenetic changes?

**Summary and General Comments**

Reviewer #1: The authors conducted a systematic analysis of the available papers on the methylation state of individual genes in human samples from patients suffered from Opisthorchis viverrini-associated cholangiocarcinoma. The study was conducted to identify individual target genes for the subsequent development of gene therapy. The materials and methods are presented in clear and straightforward style. Nevertheless, I have suggestions for the Results section that would improve the manuscript since it requires major revision.

1. The lack of line numbering is unhelpful for the reviewer

2. In my opinion, the manuscript does not have enough examples of differences in hypermethylation of regulatory regions of individual genes compared to that in other types of cancer. I recommend indicating whether this is a characteristic feature of Ov-associated CCA or not. At the same time, the authors still provide such information about individual genes (Trop2 and RIZ1), but not for all genes. If there is no such information, then I recommend mentioning that there is no information.

3. The study also lacks a table that would summarize the most important findings for the characterization of Ov-associated CCA, in particular a table with examples of specific hypermethylated regions of genes unique to Ov-CCA would be useful.

4. It is necessary to write all genes and all proteins in accordance with the nomenclature rules for writing human genes (please refer to HUGO Gene Nomenclature Committee (HGNC)

5. This review is not sufficiently illustrated. In my opinion, the manuscript lacks clear gene networks or crosstalks between signaling pathways to illustrate the main findings.

Minor 

1. page 29, 3rd paragraph – unreadable characters,

2. the Reference list is written in a different style and font 

3. p.27, mutp53 should be decrypted the first time it appears in the text

Reviewer #2: This review identified epigenetic changes and Wnt/β-catenin pathway deregulation as key drivers in Ov-CCA pathogenesis. This study underscores the importance of the epigenetic modifications in Ov-CCA development, suggesting novel therapeutic targets within disrupted signaling networks. It is very interesting and valuable for therapy of Ov-CCA.

Reviewer #3: (No Response)

PLOS authors have the option to publish the peer review history of their article (what does this mean?). If published, this will include your full peer review and any attached files.

Reviewer #1: No

Reviewer #2: No

Reviewer #3: No
---

## [Decision Letter · Decision Letter 1]

19 Aug 2024

Dear Dr. Suttiprapa,

We are pleased to inform you that your manuscript 'Epigenetic Biomarkers and the Wnt/β-Catenin Pathway in Opisthorchis viverrini-associated Cholangiocarcinoma: A Scoping Review on Therapeutic Opportunities' has been provisionally accepted for publication in PLOS Neglected Tropical Diseases.

Best regards,

Aysegul Taylan Ozkan, M.D., Ph.D.,

Academic Editor

Richard Bradbury

Section Editor

Reviewer's Responses to Questions

**Key Review Criteria Required for Acceptance?**

**Methods**

-Are the objectives of the study clearly articulated with a clear testable hypothesis stated?

-Is the study design appropriate to address the stated objectives?

-Is the population clearly described and appropriate for the hypothesis being tested?

-Is the sample size sufficient to ensure adequate power to address the hypothesis being tested?

-Were correct statistical analysis used to support conclusions?

-Are there concerns about ethical or regulatory requirements being met?

Reviewer #1: (No Response)

Reviewer #3: (No Response)

**Results**

-Does the analysis presented match the analysis plan?

-Are the results clearly and completely presented?

-Are the figures (Tables, Images) of sufficient quality for clarity?

Reviewer #1: (No Response)

Reviewer #3: (No Response)

**Conclusions**

-Are the conclusions supported by the data presented?

-Are the limitations of analysis clearly described?

-Do the authors discuss how these data can be helpful to advance our understanding of the topic under study?

-Is public health relevance addressed?

Reviewer #1: (No Response)

Reviewer #3: (No Response)

**Editorial and Data Presentation Modifications?**

Reviewer #1: (No Response)

Reviewer #3: (No Response)

**Summary and General Comments**

Reviewer #1: (No Response)

Reviewer #3: The authors have clearly explained all comments/questions. This manuscript has enough scientific merit to be published in the journal.

PLOS authors have the option to publish the peer review history of their article (what does this mean?). If published, this will include your full peer review and any attached files.

Reviewer #1: No

Reviewer #3: No

---

## [Editor Report · Acceptance letter]

1 Sep 2024

Dear Dr. Suttiprapa,

We are delighted to inform you that your manuscript, "Epigenetic Biomarkers and the Wnt/β-Catenin Pathway in Opisthorchis viverrini-associated Cholangiocarcinoma: A Scoping Review on Therapeutic Opportunities," has been formally accepted for publication in PLOS Neglected Tropical Diseases.

Best regards,

Shaden Kamhawi

co-Editor-in-Chief

Paul Brindley

co-Editor-in-Chief
